# Rethinking pooling in graph neural networks

**Diego Mesquita**[1]*, **Amauri H. Souza**[2]*, **Samuel Kaski**[1,3]
[1]Aalto University [2]Federal Institute of Ceará [3]University of Manchester
{diego.mesquita, samuel.kaski}@aalto.fi, amauriholanda@ifce.edu.br

## Abstract

Graph pooling is a central component of a myriad of graph neural network (GNN) architectures. As an inheritance from traditional CNNs, most approaches formulate graph pooling as a cluster assignment problem, extending the idea of local patches in regular grids to graphs. Despite the wide adherence to this design choice, no work has rigorously evaluated its influence on the success of GNNs. In this paper, we build upon representative GNNs and introduce variants that challenge the need for locality-preserving representations, either using randomization or clustering on the complement graph. Strikingly, our experiments demonstrate that using these variants does not result in any decrease in performance. To understand this phenomenon, we study the interplay between convolutional layers and the subsequent pooling ones. We show that the convolutions play a leading role in the learned representations. In contrast to the common belief, local pooling is not responsible for the success of GNNs on relevant and widely-used benchmarks.

## 1 Introduction

The success of graph neural networks (GNNs) [3, 36] in many domains [9, 15, 25, 41, 46, 50] is due to their ability to extract meaningful representations from graph-structured data. Similarly to convolutional neural networks (CNNs), a typical GNN sequentially combines local filtering, non-linearity, and (possibly) pooling operations to obtain refined graph representations at each layer. Whereas the convolutional filters capture local regularities in the input graph, the interleaved pooling operations reduce the graph representation while ideally preserving important structural information.

Although strategies for graph pooling come in many flavors [26, 30, 47, 49], most GNNs follow a hierarchical scheme in which the pooling regions correspond to graph clusters that, in turn, are combined to produce a coarser graph [4, 7, 13, 21, 47, 48]. Intuitively, these clusters generalize the notion of local neighborhood exploited in traditional CNNs and allow for pooling graphs of varying sizes. The cluster assignments can be obtained via deterministic clustering algorithms [4, 7] or be learned in an end-to-end fashion [21, 47]. Also, one can leverage node embeddings [21], graph topology [8], or both [47, 48], to pool graphs. We refer to these approaches as local pooling.

Together with attention-based mechanisms [24, 26], the notion that clustering is a must-have property of graph pooling has been tremendously influential, resulting in an ever-increasing number of pooling schemes [14, 18, 21, 27, 48]. Implicit in any pooling approach is the belief that the quality of the cluster assignments is crucial for GNNs performance. Nonetheless, to the best of our knowledge, this belief has not been rigorously evaluated.

Misconceptions not only hinder new advances but also may lead to unnecessary complexity and obfuscate interpretability. This is particularly critical in graph representation learning, as we have seen a clear trend towards simplified GNNs [5, 6, 11, 31, 43].

---

In this paper, we study the extent to which local pooling plays a role in GNNs. In particular, we choose representative models that are popular or claim to achieve state-of-the-art performances and simplify their pooling operators by eliminating any clustering-enforcing component. We either apply randomized cluster assignments or operate on complementary graphs. Surprisingly, the empirical results show that the non-local GNN variants exhibit comparable, if not superior, performance to the original methods in all experiments.

To understand our findings, we design new experiments to evaluate the interplay between convolutional layers and pooling; and analyze the learned embeddings. We show that graph coarsening in both the original methods and our simplifications lead to homogeneous embeddings. This is because successful GNNs usually learn low-pass filters at early convolutional stages. Consequently, the specific way in which we combine nodes for pooling becomes less relevant.

In a nutshell, the contributions of this paper are: $i$) we show that popular and modern representative GNNs do not perform better than simple baselines built upon randomization and non-local pooling; $ii$) we explain why the simplified GNNs work and analyze the conditions for this to happen; and $iii$) we discuss the overall impact of pooling in the design of efficient GNNs. Aware of common misleading evaluation protocols [10, 11], we use benchmarks on which GNNs have proven to beat structure-agnostic baselines. We believe this work presents a sanity-check for local pooling, suggesting that novel pooling schemes should count on more ablation studies to validate their effectiveness.

**Notation.**  We represent a graph $\mathcal{G}$, with $n > 0$ nodes, as an ordered pair $(\boldsymbol{A}, \boldsymbol{X})$ comprising a symmetric adjacency matrix $\boldsymbol{A} \in \{0,1\}^{n \times n}$ and a matrix of node features $\boldsymbol{X} \in \mathbb{R}^{n \times d}$. The matrix $\boldsymbol{A}$ defines the graph structure: two nodes $i, j$ are connected if and only if $A_{ij} = 1$. We denote by $\boldsymbol{D}$ the diagonal degree matrix of $\mathcal{G}$, i.e., $D_{ii} = \sum_j A_{ij}$. We denote the complement of $\mathcal{G}$ by $\bar{\mathcal{G}} = (\bar{\boldsymbol{A}}, \boldsymbol{X})$, where $\bar{\boldsymbol{A}}$ has zeros in its diagonal and $\bar{A}_{ij} = 1 - A_{ij}$ for $i \neq j$.

## 2    Exposing local pooling

### 2.1    Experimental setup

**Models.**  To investigate the relevance of local pooling, we study three representative models. We first consider GRACLUS [8], an efficient graph clustering algorithm that has been adopted as a pooling layer in modern GNNs [7, 34]. We combine GRACLUS with a sum-based convolutional operator [28]. Our second choice is the popular *differential pooling* model (DIFFPOOL) [47]. DIFFPOOL is the pioneering approach to learned pooling schemes and has served as inspiration to many methods [14]. Last, we look into the *graph memory network* (GMN) [21], a recently proposed model that reports state-of-the-art results on graph-level prediction tasks. Here, we focus on local pooling mechanisms and expect the results to be relevant for a large class of models whose principle is rooted in CNNs.

**Tasks and datasets.**  We use four graph-level prediction tasks as running examples: predicting the constrained solubility of molecules (ZINC, [20]), classifying chemical compounds regarding their activity against lung cancer (NCI1, [40]); categorizing ego-networks of actors w.r.t. the genre of the movies in which they collaborated (IMDB-B, [45]); and classifying handwritten digits (Superpixels MNIST, [1, 10]). The datasets cover graphs with none, discrete, and continuous node features. For completeness, we also report results on five other broadly used datasets in Section 3.1. Statistics of the datasets are available in the supplementary material (Section A.1).

**Evaluation.**  We split each dataset into train (80%), validation (10%) and test (10%) data. For the regression task, we use the mean absolute error (MAE) as performance metric. We report statistics of the performance metrics over 20 runs with different seeds. Similarly to the evaluation protocol in [10], we train all models with Adam [22] and apply learning rate decay, ranging from initial $10^{-3}$ down to $10^{-5}$, with decay ratio of 0.5 and patience of 10 epochs. Also, we use early stopping based on the validation accuracy. For further details, we refer to Appendix A in the supplementary material.

Notably, we do not aim to benchmark the performance of the GNN models. Rather, we want to isolate local pooling effects. Therefore, for model selection, we follow guidelines provided by the original authors or in benchmarking papers and simply modify the pooling mechanism, keeping the remaining model structure untouched. All methods were implemented in PyTorch [12, 33] and our code is available at `https://github.com/AaltoPML/Rethinking-pooling-in-GNNs`.

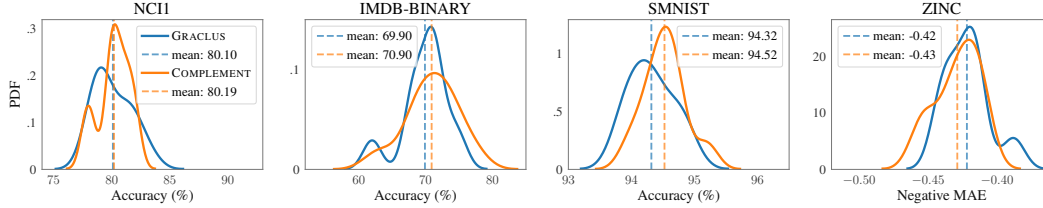

Figure 1: Comparison between GRACLUS and COMPLEMENT. The plots show the distribution of the performances over 20 runs. In all tasks, GRACLUS and COMPLEMENT achieve similar performance. The results indicate that pooling nearby nodes is not relevant to obtain a successful pooling scheme.

## Case 1: Pooling with off-the-shelf graph clustering

We first consider a network design that resembles standard CNNs. Following architectures used in [7, 12, 13], we alternate graph convolutions [28] and pooling layers based on graph clustering [8]. At each layer, a neighborhood aggregation step combines each node feature vector with the features of its neighbors in the graph. The features are linearly transformed before running through a component-wise non-linear function (e.g., ReLU). In matrix form, the convolution is

$$\boldsymbol{Z}^{(l)} = \text{ReLU}\left(\boldsymbol{X}^{(l-1)}\boldsymbol{W}_1^{(l)} + \boldsymbol{A}^{(l-1)}\boldsymbol{X}^{(l-1)}\boldsymbol{W}_2^{(l)}\right) \quad \text{with } (\boldsymbol{A}^{(0)}, \boldsymbol{X}^{(0)}) = (\boldsymbol{A}, \boldsymbol{X}), \quad (1)$$

where $\boldsymbol{W}_2^{(l)}, \boldsymbol{W}_1^{(l)} \in \mathbb{R}^{d_{l-1} \times d_l}$ are model parameters, and $d_l$ is the embedding dimension at layer $l$.

The next step consists of applying the GRACLUS algorithm [8] to obtain a cluster assignment matrix $\boldsymbol{S}^{(l)} \in \{0,1\}^{n_{l-1} \times n_l}$ mapping each node to its cluster index in $\{1, \ldots, n_l\}$, with $n_l < n_{l-1}$ clusters. We then coarsen the features by max-pooling the nodes in the same cluster:

$$X_{kj}^{(l)} = \max_{i:S_{ik}^{(l)}=1} Z_{ij}^{(l)} \qquad k = 1, \ldots, n_l \tag{2}$$

and coarsen the adjacency matrix such that $\boldsymbol{A}_{ij}^{(l)} = 1$ iff clusters $i$ and $j$ have neighbors in $\mathcal{G}^{(l-1)}$:

$$\boldsymbol{A}^{(l)} = \boldsymbol{S}^{(l)\mathsf{T}}\boldsymbol{A}^{(l-1)}\boldsymbol{S}^{(l)} \quad \text{with } A_{kk}^{(l)} = 0 \quad k = 1, \ldots, n_l. \tag{3}$$

**Clustering the complement graph.** The clustering step holds the idea that good pooling regions, equivalently to their CNN counterparts, should group nearby nodes. To challenge this intuition, we follow an opposite idea and set pooling regions by grouping nodes that are not connected in the graph. In particular, we compute the assignments $\boldsymbol{S}^{(l)}$ by applying GRACLUS to the complement graph $\bar{\mathcal{G}}^{(l-1)}$ of $\mathcal{G}^{(l-1)}$. Note that we only employ the complement graph to compute cluster assignments $\boldsymbol{S}^{(l)}$. With the assignments in hand, we apply the pooling operation (Equations 2 and 3) using the original graph structure. Henceforth, we refer to this approach as COMPLEMENT.

**Results.** Figure 1 shows the distribution of the performance of the standard approach (GRACLUS) and its variant that operates on the complement graph (COMPLEMENT). In all tasks, both models perform almost on par and the distributions have a similar shape.

Despite their simplicity, GRACLUS and COMPLEMENT are strong baselines (see Table 1). For instance, Errica et al. [11] report GIN [44] as the best performing model for the NCI1 dataset, achieving $80.0 \pm 1.4$ accuracy. This is indistinguishable from COMPLEMENT's performance ($80.1 \pm 1.6$). We observe the same trend on IMDB-B, GraphSAGE [16] obtains $69.9 \pm 4.6$ and COMPLEMENT scores

Table 1: GRACLUS and COMPLEMENT performances are similar to those from the best performing isotropic GNNs (GIN and GraphSAGE) in [11] or [10]. For ZINC, lower is better.

| Models | NCI1 ↑ | IMDB-B ↑ | SMNIST ↑ | ZINC ↓ |
|---|---|---|---|---|
| GraphSAGE | $76.0 \pm 1.8$ | $69.9 \pm 4.6$ | $97.1 \pm 0.02$ | $0.41 \pm 0.01$ |
| GIN | $80.0 \pm 1.4$ | $66.8 \pm 3.9$ | $94.0 \pm 1.30$ | $0.41 \pm 0.01$ |
| GRACLUS | $80.1 \pm 1.6$ | $69.9 \pm 3.5$ | $94.3 \pm 0.34$ | $0.42 \pm 0.02$ |
| COMPLEMENT | $80.2 \pm 1.4$ | $70.9 \pm 3.9$ | $94.5 \pm 0.33$ | $0.43 \pm 0.02$ |

$70.6 \pm 5.1$ — a less than $0.6\%$ difference. Using the same data splits as [10], COMPLEMENT and GIN perform within a margin of $1\%$ in accuracy (SMNIST) and $0.01$ in absolute error (ZINC).

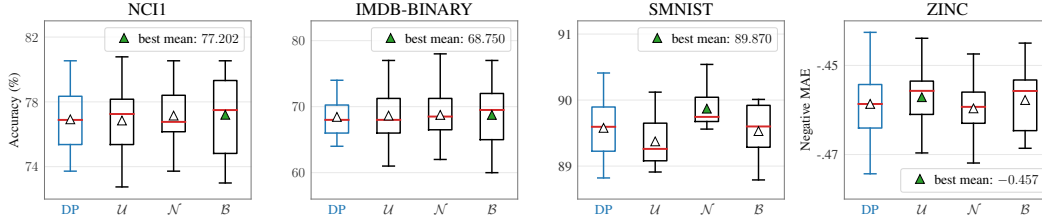

Figure 2: Boxplots for DIFFPOOL (DP) and its random variants: Uniform $\mathcal{U}(0,1)$, Normal $\mathcal{N}(0,1)$, and Bernoulli $\mathcal{B}(0.3)$. For all datasets, at least one of the random models achieves higher average accuracy than DIFFPOOL. Also, random pooling does not consistently lead to higher variance. The results show that the learned pooling assignments are not relevant for the performance of DIFFPOOL.

## Case 2: Differential pooling

DIFFPOOL [47] uses a GNN to learn cluster assignments for graph pooling. At each layer $l$, the soft cluster assignment matrix $\boldsymbol{S}^{(l)} \in \mathbb{R}^{n_{l-1} \times n_l}$ is

$$\mathbf{S}^{(l)} = \mathrm{softmax}\left(\mathrm{GNN}_1^{(l)}(\boldsymbol{A}^{(l-1)}, \boldsymbol{X}^{(l-1)})\right) \quad \text{with } (\boldsymbol{A}^{(0)}, \boldsymbol{X}^{(0)}) = (\boldsymbol{A}, \boldsymbol{X}). \tag{4}$$

The next step applies $\boldsymbol{S}^{(l)}$ and a second GNN to compute the graph representation at layer $l$:

$$\boldsymbol{X}^{(l)} = \boldsymbol{S}^{(l)\mathsf{T}}\mathrm{GNN}_2^{(l)}(\boldsymbol{A}^{(l-1)}, \boldsymbol{X}^{(l-1)}) \qquad \text{and} \qquad \boldsymbol{A}^{(l)} = \boldsymbol{S}^{(l)\mathsf{T}}\boldsymbol{A}^{(l-1)}\boldsymbol{S}^{(l)}. \tag{5}$$

During training, DIFFPOOL employs a sum of three loss functions: $i$) a supervised loss; $ii$) the Frobenius norm between $\boldsymbol{A}^{(l)}$ and the Gramian of the cluster assignments, at each layer, i.e., $\sum_l \|\boldsymbol{A}^{(l)} - \boldsymbol{S}^{(l)}\boldsymbol{S}^{(l)\mathsf{T}}\|$; $iii$) the entropy of the cluster assignments at each layer. The second loss is referred to as the *link prediction loss* and enforces nearby nodes to be pooled together. The third loss penalizes the entropy, encouraging sharp cluster assignments.

**Random assignments.** To confront the influence of the learned cluster assignments, we replace $\boldsymbol{S}^{(l)}$ in Equation 4 with a normalized random matrix $\mathrm{softmax}(\tilde{\boldsymbol{S}}^{(l)})$. We consider three distributions:

$$\text{(Uniform) } \tilde{S}_{ij}^{(l)} \sim \mathcal{U}(a,b) \qquad \text{(Normal) } \tilde{S}_{ij}^{(l)} \sim \mathcal{N}(\mu, \sigma^2) \qquad \text{(Bernoulli) } \tilde{S}_{ij}^{(l)} \sim \mathcal{B}(\alpha) \tag{6}$$

We sample the assignment matrix before training starts and do not propagate gradients through it.

**Results.** Figure 2 compares DIFFPOOL against the randomized variants. In all tasks, the highest average accuracy is due to a randomized approach. Nonetheless, there is no clear winner among all methods. Notably, the variances obtained with the random pooling schemes are not significantly higher than those from DIFFPOOL.

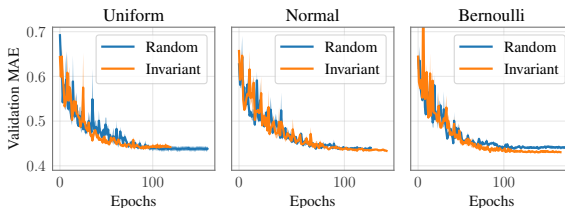

Figure 3: MAE obtained over random permutations of each graph in the validation set. Both Invariant and fully Random produce similar predictions throughout the training.

**Remark 1.** *Permutation invariance is an important property of most GNNs that assures consistency across different graph representations. However, the randomized variants break the invariance of* DIFF-POOL. *A simple fix consists of taking $\tilde{\boldsymbol{S}}^{(l)} = \boldsymbol{X}^{(l-1)}\tilde{\boldsymbol{S}}'$, where $\tilde{\boldsymbol{S}}' \in \mathbb{R}^{d_{l-1} \times n_l}$ is a random matrix. Figure 3 compares the randomized variants with and without this fix w.r.t. the validation error on artificially permuted graphs during training on the ZINC dataset. Results suggest that the variants are approximately invariant.*

## Case 3: Graph memory networks

Graph memory networks (GMNs) [21] consist of a sequence of memory layers stacked on top of a GNN, also known as the initial query network. We denote the output of the initial query network by $\boldsymbol{Q}^{(0)}$. The first step in a memory layer computes kernel matrices between input queries $\boldsymbol{Q}^{(l-1)} = [\boldsymbol{q}_1^{(l-1)}, \ldots, \boldsymbol{q}_{n_{l-1}}^{(l-1)}]^{\intercal}$ and multi-head keys $\boldsymbol{K}_h^{(l)} = [\boldsymbol{k}_{1h}^{(l)}, \ldots, \boldsymbol{k}_{n_l h}^{(l)}]^{\intercal}$:

$$\boldsymbol{S}_h^{(l)} : S_{ijh}^{(l)} \propto \left(1 + \|\boldsymbol{q}_i^{(l-1)} - \boldsymbol{k}_{jh}^{(l)}\|^2/\tau\right)^{-\frac{\tau+1}{2}} \quad \forall h = 1 \ldots H, \tag{7}$$

where $H$ is the number of heads and $\tau$ is the degrees of freedom of the Student's $t$-kernel.

We then aggregate the multi-head assignments $\boldsymbol{S}_h^{(l)}$ into a single matrix $\boldsymbol{S}^{(l)}$ using a $1{\times}1$ convolution followed by row-wise softmax normalization. Finally, we pool the node embeddings $\boldsymbol{Q}^{(l-1)}$ according to their soft assignments $\boldsymbol{S}^{(l)}$ and apply a single-layer neural net:

$$\boldsymbol{Q}^{(l)} = \mathrm{ReLU}\left(\boldsymbol{S}^{(l)\intercal} \boldsymbol{Q}^{(l-1)} \boldsymbol{W}^{(l)}\right). \tag{8}$$

In this notation, queries $\boldsymbol{Q}^{(l)}$ correspond to node embeddings $\boldsymbol{X}^{(l)}$ for $l > 0$. Also, note that the memory layer does not leverage graph structure information as it is fully condensed into $\boldsymbol{Q}^{(0)}$. Following [21], we use a GNN as query network. In particular, we employ a two layer network with the same convolutional operator as in Equation 1.

The loss function employed to learn GMNs consists of a convex combination of: $i$) a supervised loss; and $ii$) the Kullback-Leibler divergence between the learned assignments and their self-normalized squares. The latter aim to enforce sharp soft-assignments, similarly to the entropy loss in DIFFPOOL.

**Remark 2.** *Curiously, the intuition behind the loss that allegedly improves cluster purity might be misleading. For instance, uniform cluster assignments, the ones the loss was engineered to avoid, are a perfect minimizer for it. We provide more details in Appendix D.*

**Simplifying GMN.** The principle behind GMNs consists of grouping nodes based on their similarities to learned keys (centroids). To scrutinize this principle, we propose two variants. In the first, we replace the kernel in Equation 7 by the euclidean distance taken from fixed keys drawn from a uniform distribution. Opposite to vanilla GMNs, the resulting assignments group nodes that are farthest from a key. The second variant substitutes multi-head assignment matrices for a fixed matrix whose entries are independently sampled from a uniform distribution. We refer to these approaches, respectively, as DISTANCE and RANDOM.

**Results.** Figure 4 compares GMN with its simplified variants. For all datasets, DISTANCE and RANDOM perform on par with GMN, with slightly better MAE for the ZINC dataset. Also, the variants present no noticeable increase in variance.

It is worth mentioning that the simplified GMNs are naturally faster to train as they have significantly less learned parameters. In the case of DISTANCE, keys are taken as constants once sampled. Additionally, RANDOM bypasses the computation of the pairwise distances in Equation 7, which dominates the time of the forward pass in GMNs. On the largest dataset (SMNIST), DISTANCE takes up to half the time of GMN (per epoch), whereas RANDOM is up to ten times faster than GMN.

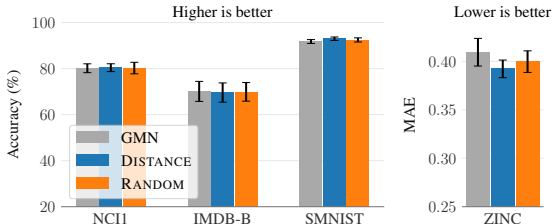

Figure 4: GMN versus its randomized variants DISTANCE and RANDOM. The plots show that pooling based on dissimilarity measure (distance) instead of similarity (kernel) does not have a negative impact in performance. The same holds true when we employ random assignments. Both variants achieve lower MAE for the regression task (ZINC).

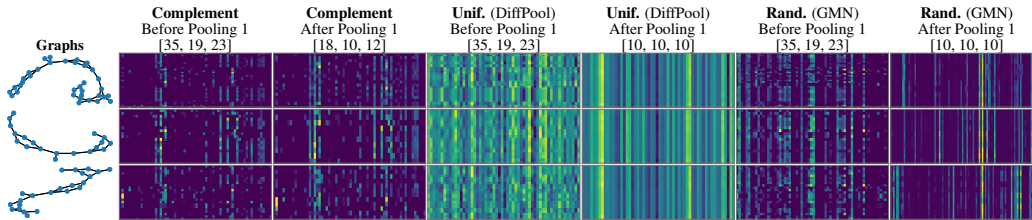

Figure 5: **Non-local pooling**: activations for three arbitrary graphs, before and after the first pooling layer. The convolutional layers learn to output similar embeddings within a same graph. As a result, methods become approximately invariant to how embeddings are pooled. Darker tones denote lower activation values. To better assess homogeneity, we normalize the color range of each embedding matrix individually. The number of nodes for each embedding matrix is given inside brackets.

# 3   Analysis

The results in the previous section are counter-intuitive. We now analyze the factors that led to these results. We show that the convolutions preceding the pooling layers learn representations which are approximately invariant to how nodes are grouped. In particular, GNNs learn smooth node representations at the early layers. To experimentally show this, we remove the initial convolutions that perform this early smoothing. As a result, all networks experience a significant drop in performance. Finally, we show that local pooling offers no benefit in terms of accuracy to the evaluated models.

**Pooling learns approximately homogeneous node representations.**   The results with the random pooling methods suggest that any convex combination of the node features enables us to extract good graph representation. Intuitively, this is possible if the nodes display similar activation patterns before pooling. If we interpret convolutions as filters defined over node features/signals, this phenomenon can be explained if the initial convolutional layers extract low-frequency information across the graph input channels/embedding components. To evaluate this intuition, we compute the activations before the first pooling layer and after each of the subsequent pooling layers. Figure 5 shows activations for the random pooling variants of DIFFPOOL and GMN, and the COMPLEMENT approach on ZINC.

The plots in Figure 5 validate that the first convolutional layers produce node features which are relatively homogeneous within the same graph, specially for the randomized variants. The networks learn features that resemble vertical bars. As expected, the pooling layers accentuate this phenomenon, extracting even more similar representations. We report embeddings for other datasets in Appendix E.

Even methods based on local pooling tend to learn homogeneous representations. As one can notice from Figure 6, DIFFPOOL and GMN show smoothed patterns in the outputs of their initial pooling layer. This phenomenon explains why the performance of the randomized approaches matches those of their original counterparts. The results suggest that the loss terms designed to enforce local clustering are either not beneficial to the learning task or are obfuscated by the supervised loss. This observation does not apply to GRACLUS, as it employs a deterministic clustering algorithm, separating the possibly competing goals of learning hierarchical structures and minimizing a supervised loss.

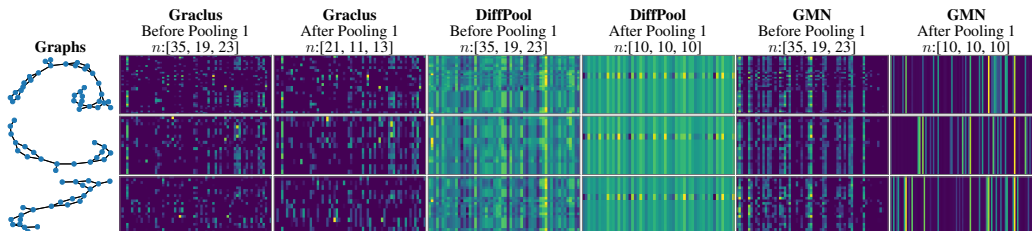

Figure 6: **Local pooling**: activations for the same graphs from Figure 5, before and after the first pooling layer. Similar to the simplified variants, their standard counterparts learn homogenous embeddings, specially for DIFFPOOL and GMN. The low variance across node embedding columns illustrates this homogeneity. Darker tones denote lower values and colors are not comparable across embedding matrices. Brackets (top) show the number of nodes after each layer for each graph.

To further gauge the impact of the unsupervised loss on the performance of these GNNs, we compare two DIFFPOOL models trained with the link prediction loss multiplied by the weighting factors $\lambda = 10^0$ and $\lambda = 10^3$. Figure 7 shows the validation curves of the supervised loss for ZINC and SMNIST. We observe that the supervised losses for models with both of the $\lambda$ values converge to a similar point, at a similar rate. This validates that the un-

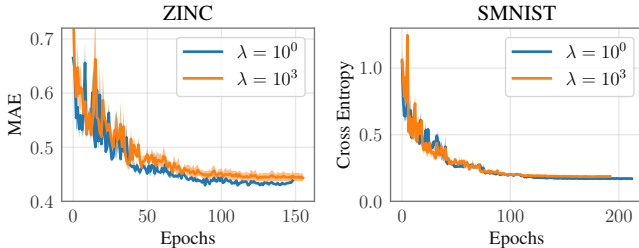

Figure 7: Supervised loss on validation set for ZINC and SMNIST datasets. Scaling up the unsupervised loss has no positive impact on the performance of DIFFPOOL.

supervised loss (link prediction) has little to no effect on the predictive performance of DIFFPOOL. We have observed a similar behavior for GMNs, which we report in the supplementary material.

**Discouraging smoothness.** In Figure 6, we observe homogeneous node representations even before the first pooling layer. This naturally poses a challenge for the upcoming pooling layers to learn meaningful local structures. These homogeneous embeddings correspond to low frequency signals defined on the graph nodes. In the general case, achieving such patterns is only possible with more than a single convolution. This can be explained from a spectral perspective. Since each convolutional layer corresponds to filters that act linearly in the spectral domain, a single convolution cannot filter out specific frequency bands. These ideas have already been exploited to develop simplified GNNs [31, 43] that compute fixed polynomial filters in a normalized spectrum.

Remarkably, using multiple convolutions to obtain initial embeddings is common practice [18, 26]. To evaluate its impact, we apply a single convolution before the first pooling layer in all networks. We then compare these networks against the original implementations. Table 2 displays the results. All models report a significant drop in performance with a single convolutional layer. On NCI1, the methods obtain accura-

Table 2: All networks experience a decrease in performance when implemented with a single convolution before pooling (numbers in red). For ZINC, lower is better.

| # Convolutions | NCI1 $= 1$ | NCI1 $> 1$ | IMDB-B $= 1$ | IMDB-B $> 1$ | SMNIST $= 1$ | SMNIST $> 1$ | ZINC $= 1$ | ZINC $> 1$ |
|---|---|---|---|---|---|---|---|---|
| GRACLUS | 76.6 | 80.1 | 69.7 | 69.9 | 89.1 | 94.3 | 0.458 | 0.429 |
| COMPLEMENT | 74.2 | 80.2 | 69.8 | 70.9 | 81.0 | 94.5 | 0.489 | 0.431 |
| DIFFPOOL | 71.3 | 76.9 | 68.5 | 68.5 | 66.6 | 89.6 | 0.560 | 0.459 |
| $\mathcal{U}$ DIFFPOOL | 72.8 | 76.8 | 69.0 | 68.7 | 66.4 | 89.4 | 0.544 | 0.457 |
| GMN | 77.1 | 80.2 | 68.9 | 69.9 | 87.4 | 92.4 | 0.469 | 0.409 |
| Rand. GMN | 76.4 | 80.2 | 66.6 | 66.8 | 90.0 | 94.0 | 0.465 | 0.400 |

cies about 4% lower, on average. Likewise, GMN and GRACLUS report a 4% performance drop on SMNIST. With a single initial GraphSAGE convolution, the performances of Diffpool and its uniform variant become as lower as 66.4% on SMNIST. This dramatic drop is not observed only for IMDB-B, which counts on constant node features and therefore may not benefit from the additional convolutions. Note that under reduced expressiveness, pooling far away nodes seems to impose a negative inductive bias as COMPLEMENT consistently fails to rival the performance of GRACLUS. Intuitively, the number of convolutions needed to achieve smooth representations depend on the richness of the initial node features. Many popular datasets rely on one-hot features and might require a small number of initial convolutions. Extricating these effects is outside the scope of this work.

**Is pooling overrated?** Our experiments suggest that local pooling does not play an important role on the performance of GNNs. A natural next question is whether local pooling presents any advantage over global pooling strategies. We run a GNN with global mean pooling on top of three convolutional layers, the results are: $79.65 \pm 2.07$ (NCI), $71.05 \pm 4.58$ (IMDB-B), $95.20 \pm 0.18$ (SMNIST), and $0.443 \pm 0.03$ (ZINC). These performances are on par with our previous results obtained using GRACLUS, DIFFPOOL and GMN.

Regardless of how surprising this may sound, our findings are consistent with results reported in the literature. For instance, GraphSAGE networks outperform DIFFPOOL in a rigorous evaluation protocol [10, 11], although GraphSAGE is the convolutional component of DIFFPOOL. Likewise, but in the context of attention, Knyazev et al. [24] argue that, except under certain circumstances, general attention mechanisms are negligible or even harmful. Similar findings also hold for CNNs [35, 39].

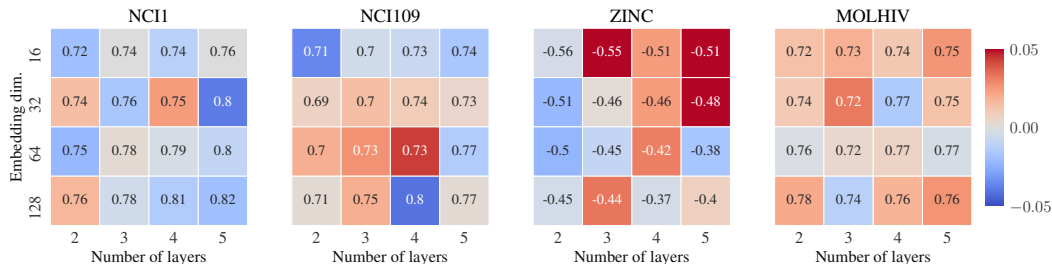

Figure 8: Performance gap between GRACLUS and COMPLEMENT for different hyperparameter settings. Numbers denote the performance of GRACLUS — negative MAE for ZINC, AUROC for MOLHIV, and accuracy for the rest. Red tones denote cases on which COMPLEMENT obtains better performance. For all datasets, the performance gap is not larger than 0.05 (or 5% in accuracy). The predominant light tones demonstrate that both models obtain overall similar results, with no clear trend toward specific hyperparameter choices.

**Permutation invariance** is a relevant property for GNNs as it guarantees that the network's predictions do not vary with the graph representation. For completeness, we state in Appendix C the permutation invariance of the simplified GNNs discussed in Section 2.

### 3.1 Additional results

**More datasets.** We consider four additional datasets commonly used to assess GNNs, three of which are part of the TU datasets [29]: PROTEINS, NCI109, and DD; and one from the recently proposed Open Graph Benchamark (OGB) framework [17]: MOLHIV. Since Ivanov et al. [19] showed that the IMDB-B dataset is affected by a serious isomorphism bias, we also report results on the "cleaned" version of the IMDB-B dataset, hereafter refereed to as IMDB-C.

Table 3 shows the results for the additional datasets. Similarly to what we have observed in the previous experiments, non-local pooling performs on par with local pooling. The largest performance gap occurs for the PROTEINS and MOHIV datasets, on which COMPLEMENT achieves accuracy around 3% higher than GRACLUS, on average. The performance of all methods on IMDB-C does not differ significantly from their performance on IMDB-B. However, removing the isomorphisms in IMDB-B reduces the dataset size, which clearly increases variance.

Table 3: Results for the additional datasets. Again, we observe that non-local pooling yields results as good as local pooling.

| Models | IMDB-C | PROTEINS | NCI109 | DD | MOLHIV |
|---|---|---|---|---|---|
| GRACLUS | $70.5 \pm 8.3$ | $72.6 \pm 4.1$ | $77.7 \pm 2.6$ | $72.1 \pm 5.2$ | $74.6 \pm 1.8$ |
| COMPLEMENT | $70.1 \pm 7.0$ | $75.3 \pm 3.1$ | $77.0 \pm 2.4$ | $72.1 \pm 3.6$ | $76.4 \pm 1.9$ |
| DIFFPOOL | $70.9 \pm 6.7$ | $75.2 \pm 4.0$ | $72.2 \pm 1.5$ | $76.9 \pm 4.4$ | $75.1 \pm 1.3$ |
| $\mathcal{N}$-DIFFPOOL | $70.3 \pm 7.5$ | $74.6 \pm 3.8$ | $71.8 \pm 2.1$ | $78.3 \pm 4.1$ | $74.8 \pm 1.3$ |
| GMN | $68.4 \pm 7.1$ | $74.8 \pm 3.3$ | $77.0 \pm 1.7$ | $73.9 \pm 4.6$ | $73.7 \pm 1.8$ |
| Random-GMN | $70.2 \pm 8.0$ | $74.8 \pm 3.9$ | $76.9 \pm 2.0$ | $74.3 \pm 3.6$ | $73.5 \pm 2.6$ |

**Another pooling method.** We also provide results for MINCUTPOOL [2], a recently proposed pooling scheme based on spectral clustering. This scheme integrates an unsupervised loss which stems from a relaxation of a MINCUT objective and learns to assign clusters in a spectrum-free way. We compare MINCUTPOOL with a random variant for all datasets employed in this work. Again, we find that a random pooling mechanism achieves comparable results to its local counterpart. Details are given in Appendix B.

**Sensitivity to hyperparameters.** As mentioned in Section 2.1, this paper does not intend to benchmark the predictive performance of models equipped with different pooling strategies. Consequently, we did not exhaustively optimize model hyperparameters. One may wonder whether our results hold for a broader set of hyperparameter choices. Figure 8 depicts the performance gap between GRACLUS and COMPLEMENT for a varying number of pooling layers and embedding dimensionality over a single run. We observe the greatest performance gap in favor of COMPLEMENT (e.g., 5 layers and 32 dimensions on ZINC), which does not amount to an improvement greater than 0.05 in MAE. Overall, we find that the choice of hyperparameters does not significantly increase the performance gap between GRACLUS and COMPLEMENT.

# 4 Related works

**Graph pooling**   usually falls into global and hierarchical approaches. Global methods aggregate the node representations either via simple flattening procedures such as summing (or averaging) the node embeddings [44] or more sophisticated set aggregation schemes [32, 49]. On the other hand, hierarchical methods [7, 14, 18, 21, 26, 27, 47] sequentially coarsen graph representations over the network's layers. Notably, Knyazev et al. [24] provides a unified view of local pooling and node attention mechanisms, and study the ability of pooling methods to generalize to larger and noisy graphs. Also, they show that the effect of attention is not crucial and sometimes can be harmful, and propose a weakly-supervised approach.

**Simple GNNs.**   Over the last years, we have seen a surge of simplified GNNs. Wu et al. [43] show that removing the nonlinearity in GCNs [23] does not negatively impact on their performances on node classification tasks. The resulting feature extractor consists of a low-pass-type filter. In [31], the authors take this idea further and evaluate the resilience to feature noise and provide insights on GCN-based designs. For graph classification, Chen et al. [6] report that linear convolutional filters followed by nonlinear set functions achieve competitive performances against modern GNNs. Cai and Wang [5] propose a strong baseline based on local node statistics for non-attributed graph classification tasks.

**Benchmarking GNNs.**   Errica et al. [11] demonstrate how the lack of rigorous evaluation protocols affects reproducibility and hinders new advances in the field. They found that structure-agnostic baselines outperform popular GNNs on at least three commonly used chemical datasets. At the rescue of GNNs, Dwivedi et al. [10] argue that this lack of proper evaluation comes mainly from using small datasets. To tackle this, they introduce a new benchmarking framework with datasets that are large enough for researchers to identify key design principles and assess statistically relevant differences between GNN architectures. Similar issues related to the use of small datasets are reported in [37].

**Understanding pooling and attention in regular domains.**   In the context of CNNs for object recognition tasks, Ruderman et al. [35] evaluate the role of pooling in CNNs w.r.t. the ability to handle deformation stability. The results show that pooling is not necessary for appropriate deformation stability. The explanation lies in the network's ability to learn smooth filters across the layers. Sheng et al. [38] and Zhao et al. [51] propose random pooling as a fast alternative to conventional CNNs. Regarding attention-based models, Wiegreffe and Pinter [42] show that attention weights usually do not provide meaningful explanations for predictions. These works demonstrate the importance of proper assessment of core assumptions in deep learning.

# 5 Conclusion

In contrast to the ever-increasing influx of GNN architectures, very few works rigorously assess which design choices are crucial for efficient learning. Consequently, misconceptions can be widely spread, influencing the development of models drinking from flawed intuitions. In this paper, we study the role of local pooling and its impact on the performance of GNNs. We show that most GNN architectures employ convolutions that can quickly lead to smooth node representations. As a result, the pooling layers become approximately invariant to specific cluster assignments. We also found that clustering-enforcing regularization is usually innocuous. In a series of experiments on accredited benchmarks, we show that extracting local information is not a necessary principle for efficient pooling. By shedding new light onto the role of pooling, we hope to contribute to the community in at least two ways: $i$) providing a simple sanity-check for novel pooling strategies; $ii$) deconstruct misconceptions and wrong intuitions related to the benefits of graph pooling.

## Acknowledgments and Disclosure of Funding

This work was funded by the Academy of Finland (Flagship programme: Finnish Center for Artificial Intelligence, FCAI, and grants 294238, 292334 and 319264). We acknowledge the computational resources provided by the Aalto Science-IT Project.

## Broader impact

Graph neural networks (GNNs) have become the *de facto* learning tools in many valuable domains such as social network analysis, drug discovery, recommender systems, and natural language processing. Nonetheless, the fundamental design principles behind the success of GNNs are only partially understood. This work takes a step further in understanding local pooling, one of the core design choices in many GNN architectures. We believe this work will help researchers and practitioners better choose in which directions to employ their time and resources to build more accurate GNNs.

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
