[Supplementary Material]

# Rethinking pooling in graph neural networks
## — Supplementary material —

## A  Implementation details

### A.1  Datasets

Table S1 reports summary statistics of the datasets used in this paper. For SMNIST and ZINC, we use the same pre-processing steps and data splits as in [10]. For the IMDB-B dataset, we use uninformative features (vector of ones) for all nodes. NCI1 and IMDB-B are part of the TU Datasets[2], a vast collection of datasets commonly used for evaluating graph kernel methods and GNNs. Errica et al. [11] show that drawing conclusions based on some of these datasets can be problematic as structure-agnostic baselines achieve higher performance than traditional GNNs. However, in their assessment, NCI1 is the only chemical dataset on which GNNs beat baselines. Also, among the social datasets, IMDB-B produces the greatest performance gap between the baseline and DiffPool ($\approx 20\%$). Table S1 also shows statistics for PROTEINS, NCI109, DD, and MOLHIV.

Table S1: Statistics of the datasets.

| Dataset | #graphs | #classes | #node feat. | #node labels | Avg #nodes | Avg #edges |
|---------|---------|----------|-------------|--------------|------------|------------|
| NCI1 | 4110 | 2 | - | 37 | 29.87 | 32.30 |
| IMDB-B | 1000 | 2 | - | - | 19.77 | 96.53 |
| SMNIST | 70000 | 10 | 3 | - | 70.57 | 564.53 |
| ZINC | 12000 | - | - | 28 | 23.16 | 49.83 |
| PROTEINS | 1113 | 2 | - | 3 | 39.06 | 72.82 |
| NCI109 | 4127 | 2 | - | 38 | 29.68 | 32.13 |
| DD | 1178 | 2 | - | 82 | 284.32 | 715.66 |
| MOLHIV | 41127 | 2 | - | 9 | 25.5 | 27.5 |

### A.2  Models

We implement all models using the PyTorch Geometric Library [12]. We apply the Adam optimizer [22] with learning rate decaying from $10^{-3}$ to $10^{-5}$. If the validation performance does not improve over 10 epochs we reduce the learning rate by half. Also, we use early stopping with patience of 50 epochs.

For the MOLHIV dataset, we propagate the edge features through a linear layer and incorporate the result in the messages of the first convolutional layer. More specifically, we add the edge embeddings to the node features and apply a ReLU function. We follow closely the procedure applied to the GIN and GCN baselines[3] for the MOLHIV dataset.

GRACLUS.  Our GRACLUS model is based on the implementation available in PyTorch Geometric. For all datasets, we employ an initial convolution to extract node embeddings. Then, we interleave convolutions and pooling layers. For instance, a 3-layer GRACLUS model consists of $conv \rightarrow conv/pool \rightarrow conv/pool \rightarrow readout \rightarrow MLP$. In all experiments, we apply global mean pooling as readout layer. Also, we adopt an MLP with a single hidden layer and the same number of hidden components as the GNN layers. The Complement model sticks to exactly the same setup. We report the specific hyperparameter values for each dataset in Table S2.

Table S2: Hyperparameters for the GRACLUS/COMPLEMENT models.

| Dataset | #Layers | |Hidden dim.| | |Batch| |
|---|---|---|---|
| NCI1 | 3 | 64 | 8 |
| IMDB-B | 2 | 64 | 8 |
| SMNIST | 2 | 64 | 8 |
| ZINC | 3 | 64 | 8 |
| PROTEINS | 3 | 64 | 64 |
| NCI109 | 3 | 64 | 64 |
| DD | 3 | 64 | 64 |
| MOLHIV | 2 | 128 | 64 |

**DIFFPOOL.** We follow the setup in [10] and use 3 GraphSAGE convolutions before and after the pooling layers, except for the first that adopts a single convolution. The embedding and pooling GNNs consist of a 1-hop GraphSAGE convolution. We apply residual connections and use embedding dimension of 32 for NCI1, SMNIST, and IMDB-B, and 56 for ZINC (similar to [10]). We use mini-batches of size 64 for SMNIST and ZINC. We report in Table S3 the list of the main hyperparameters used in the experiments. Further details can be found in our official repository. Since the experiments with DIFFPOOL have been removed from [10] in the latest version of the paper, we report results with a different architecture in Appendix B.

Table S3: Hyperparameters for DIFFPOOL and its variants.

| Dataset | #Pool. Layers | |GNN hidden dim.| | |MLP hidden dim.| | |Batch| |
|---|---|---|---|---|
| NCI1 | 1 | 32 | 32 | 16 |
| IMDB-B | 1 | 32 | 50 | 8 |
| SMNIST | 1 | 32 | 50 | 64 |
| ZINC | 1 | 56 | 56 | 64 |
| PROTEINS | 2 | 32 | 50 | 8 |
| NCI109 | 1 | 32 | 32 | 16 |
| DD | 2 | 32 | 50 | 8 |
| MOLHIV | 2 | 32 | 50 | 32 |

**GMN.** Based on the official code repository, we re-implemented the GMN using PyTorch Geometric. The initial query network consists of two basic convolutions — see Equation 1 — followed by batch normalization. We found that this basic convolution produces better results than the random walk with restart (RWR) and graph attention networks as suggested in [21]. Our design choice is also less dependent on dataset specificities, as applying RWR to graphs with multiple connected components (such as the ones on NCI1) requires additional pre-processing, for example. Table S4 shows more details on the hyperparameters used for the memory layers. We feed the output of the last memory layer to a single-hidden layer feed-forward net with fifty hidden nodes.

Table S4: Hyperparameters for the GMN models.

| Dataset | #Keys | #Heads | #Layers | |Hidden dim.| | |Batch| |
|---|---|---|---|---|---|
| NCI1 | [10, 1] | 5 | 2 | 100 | 128 |
| IMDB-B | [32, 1] | 1 | 2 | 16 | 128 |
| SMNIST | [32, 1] | 10 | 2 | 16 | 128 |
| ZINC | [10, 1] | 5 | 2 | 100 | 128 |
| PROTEINS | [32, 1] | 5 | 2 | 16 | 128 |
| NCI109 | [32, 1] | 5 | 2 | 16 | 128 |
| DD | [32, 8, 1] | 5 | 3 | 64 | 128 |
| MOLHIV | [32, 1] | 5 | 2 | 16 | 128 |

# B Additional experiments

For completeness, in this section, we report additional results using one more pooling method. We also present the results of a second version of DIFFPOOL that employs 3 graph convolutions at each pooling layer. Furthermore, we gauge the impact of the unsupervised loss of GMNs.

**MINCUTPOOL.** Bianchi et al. [2] propose a pooling scheme based on a spectrum-free formulation of spectral clustering. Similarly to DIFFPOOL, MINCUTPOOL leverages node features and graph topology to learn cluster assignments. In particular, these assignments are computed using the composition of an MLP and a GNN layer. The parameters in a MINCUTPOOL layer are learned by minimizing a minCUT objective and a supervised loss. As for the code, we use the implementation available in PyTorch Geometric. In the experiments, we apply three layers of interleaved convolution and pooling operators, as originally employed in [2].

**Randomized MINCUTPOOL.** To evaluate the effectiveness of the MINCUTPOOL layers, we follow a randomized approach. Similarly to the DIFFPOOL variants, we simply replace the learned assignment matrix with a random one, sampled from a standard normal distribution. We only apply the supervised loss function. In the following, we refer to this approach as $\mathcal{N}$-MINCUTPOOL.

**Another version of DIFFPOOL.** Here we consider the architecture choice and implementation given in [11]. Each pooling and embedding GNN encompasses a 3-layer SAGE convolution. After the final pooling layer, another group of 3-layer SAGE convolutions is used before the readout layer. We evaluate a randomized variant with normal distribution, referred to as $\mathcal{N}$-DIFFPOOL v2.

**Results.** Table S5 displays the results for MINCUTPOOL and DIFFPOOL v2. Overall, we find that the variants and the original pooling methods obtain similar performance. Notably, SMNIST stands out as an exception, as MINCUTPOOL performs significantly better (89.3%) than its randomized version (82.0%). We argue that this is because we only use one single convolution before the first pooling layer, which is consistent with the findings in Table 2. The fact that the initial features are dense might make learning smooth features harder. Nonetheless, as we have previously seen, using as few as two initial convolutions suffices to settle this performance gap on SMNIST.

Table S5: Additional results for MINCUTPOOL and a second implementation of DIFFPOOL.

| Models | NCI1 ↑ | IMDB-B ↑ | SMNIST ↑ | ZINC ↓ | PROTEINS ↑ | NCI109 ↑ | DD ↑ | MOLHIV ↑ |
|---|---|---|---|---|---|---|---|---|
| MINCUTPOOL | $76.1 \pm 1.9$ | $68.8 \pm 4.6$ | $89.3 \pm 1.0$ | $0.47 \pm 0.01$ | $75.0 \pm 3.7$ | $74.3 \pm 2.2$ | $77.2 \pm 4.1$ | $71.9 \pm 1.6$ |
| $\mathcal{N}$-MINCUTPOOL | $76.6 \pm 1.6$ | $69.7 \pm 4.8$ | $82.0 \pm 1.6$ | $0.47 \pm 0.01$ | $75.2 \pm 3.5$ | $74.7 \pm 2.1$ | $77.0 \pm 3.2$ | $72.6 \pm 2.2$ |
| DIFFPOOL v2 | $77.4 \pm 2.0$ | $67.7 \pm 3.1$ | $90.7 \pm 0.3$ | $0.50 \pm 0.01$ | $73.2 \pm 3.5$ | $75.2 \pm 1.4$ | $76.5 \pm 2.6$ | $71.8 \pm 2.2$ |
| $\mathcal{N}$-DIFFPOOL v2 | $77.4 \pm 1.4$ | $68.2 \pm 2.8$ | $89.0 \pm 0.5$ | $0.49 \pm 0.01$ | $74.6 \pm 4.0$ | $75.2 \pm 1.7$ | $75.3 \pm 3.4$ | $74.4 \pm 3.0$ |

**Unsupervised loss of GMNs.** GMNs are trained by alternating between the supervised loss and the unsupervised (purity enforcing) loss. When using the unsupervised loss, we only update the model keys; and when using the supervised loss, we update all remaining parameters. In its official implementation, GMN updates its key locations once each $p = 5$ epochs. Figure S1 shows the supervised loss on the validation set, with $p \in \{2, 5\}$, for ZINC and IMDB. As is the case for DIFFPOOL, enforcing the role of the unsupervised loss does not improve the performance of GMNs.

# C Permutation Invariance

Permutation invariance is an important property for GNNs since it guarantees that the network's predictions do not vary with the graph representation. Here we state the permutation invariance of some of the simplified GNNs discussed in Section 2. For completeness, we bring here definitions of invariant and equivariant functions over graphs.

**Definition 1** (Permutation matrix). $\boldsymbol{P}_n \in \{0,1\}^{n \times n}$ *is a permutation matrix of size $n$ if $\sum_i P_{i,j} = 1 \ \forall j$ and $\sum_j P_{i,j} = 1 \ \forall i$.*

We denote by $\mathfrak{G}$ the set of undirected graphs and by $\mathfrak{G}_n$ the set of all graphs with exactly $n$ nodes.

Figure S1: Supervised loss on validation set for ZINC and IMDB-BINARY datasets. Updating the keys using the unsupervised loss more frequently shows no positive impact on the performance of GMN. For IMDB-BINARY, updating the keys more frequently leads to a clearly worse minimum. Curves are averaged over twenty repetitions with different random seeds.

**Definition 2** (Equivariant graph function). *A function $f : \mathfrak{G}' \subseteq \mathfrak{G} \to \mathbb{R}^{n \times d}$ is equivariant if $f(\boldsymbol{P}_n \boldsymbol{A} \boldsymbol{P}_n^\intercal, \boldsymbol{P}_n \boldsymbol{X}) = \boldsymbol{P}_n f(\boldsymbol{A}, \boldsymbol{X})$ for any $\mathcal{G} = (\boldsymbol{A}, \boldsymbol{X}) \in \mathfrak{G}'$ with $n$ nodes and any permutation matrix $\boldsymbol{P}_n$, where $n$ is the number of nodes in $\mathcal{G}$.*

**Definition 3** (Invariant graph function). *Let $d > 0$. A function $f : \mathfrak{G}' \subseteq \mathfrak{G} \to \mathcal{I}$ is invariant if $f(\boldsymbol{P}_n \boldsymbol{A} \boldsymbol{P}_n^\intercal, \boldsymbol{P}_n \boldsymbol{X}) = f(\boldsymbol{A}, \boldsymbol{X})$ for any $\mathcal{G} = (\boldsymbol{A}, \boldsymbol{X}) \in \mathfrak{G}'$ and any permutation matrix $\boldsymbol{P}_n$, where $n$ is the number of nodes in $\mathcal{G}$.*

## C.1 GRACLUS

**Remark 3** (On the invariance of GRACLUS and COMPLEMENT.). *In general, the invariance of pooling methods that rely on graph clustering depends on the invariance of the clustering algorithm itself. For any invariant clustering algorithm, COMPLEMENT (see Section 2) is also invariant. However, this is naturally not the case of GRACLUS as it employs a heuristic that can return different graph cuts depending on initialization.*

## C.2 DIFFPOOL

**Theorem 1** (Invariance of randomized DIFFPOOL). *The DIFFPOOL variant in Remark 1 is invariant.*

We prove this by showing that each modified pooling layer is individually invariant. Here, we treat the $l$-th pooling layer as a function $f_l : \mathfrak{G} \to \mathfrak{G}$. Let $\mathcal{G} = \left( \boldsymbol{X}^{(l-1)}, \boldsymbol{A}^{(l-1)} \right)$ be a graph with $n_{l-1}$ nodes and $\boldsymbol{P}_{n_{l-1}}$ be a permutation matrix. Furthermore, we define:

$$\left( \boldsymbol{A}^{(l)}, \boldsymbol{X}^{(l)} \right) \coloneqq f_l \left( \boldsymbol{A}^{(l-1)}, \boldsymbol{X}^{(l-1)} \right) \text{ and } \left( \boldsymbol{A}^{(l)\prime}, \boldsymbol{X}^{(l)\prime} \right) \coloneqq f_l \left( \boldsymbol{P}_{n_{l-1}} \boldsymbol{A}^{(l-1)} \boldsymbol{P}_{n_{l-1}}^\intercal, \boldsymbol{P}_{n_{l-1}} \boldsymbol{X}^{(l-1)} \right),$$

so that our job reduces to proving $\left( \boldsymbol{A}^{(l)}, \boldsymbol{X}^{(l)} \right) = \left( \boldsymbol{A}^{(l)\prime}, \boldsymbol{X}^{(l)\prime} \right)$.

Note that modified assignment matrix for a graph with node features $\boldsymbol{X}^{(l-1)}$ is $\tilde{\boldsymbol{S}}^{(l)} = \boldsymbol{X}^{(l-1)} \tilde{\boldsymbol{S}}'$, and thus the assignment matrix for $\boldsymbol{P}_{n_{l-1}} \boldsymbol{X}^{(l-1)}$ is $\boldsymbol{P}_{n_{l-1}} \tilde{\boldsymbol{S}}^{(l)}$. Then, it follows that:

$$\begin{aligned} \boldsymbol{X}^{(l)\prime} &= \left( \boldsymbol{P}_{n_{l-1}} \tilde{\boldsymbol{S}}^{(l)} \right)^\intercal \mathrm{GNN}_2^{(l)} (\boldsymbol{P}_{n_{l-1}} \boldsymbol{A}^{(l-1)} \boldsymbol{P}_{n_{l-1}}^\intercal, \boldsymbol{P}_{n_{l-1}} \boldsymbol{X}^{(l-1)}) \\ &= \tilde{\boldsymbol{S}}^{(l)\intercal} \boldsymbol{P}_{n_{l-1}}^\intercal \boldsymbol{P}_{n_{l-1}} \mathrm{GNN}_2^{(l)} (\boldsymbol{A}^{(l-1)}, \boldsymbol{X}^{(l-1)}) \\ &= \tilde{\boldsymbol{S}}^{(l)\intercal} \mathrm{GNN}_2^{(l)} (\boldsymbol{A}^{(l-1)}, \boldsymbol{X}^{(l-1)}) \\ &= \boldsymbol{X}^{(l)} \end{aligned}$$

and for the coarsened adjacency matrix:

$$\begin{aligned} \boldsymbol{A}^{(l)\prime} &= \left( \boldsymbol{P}_{n_{l-1}} \tilde{\boldsymbol{S}}^{(l)} \right)^\intercal \boldsymbol{P}_{n_{l-1}} \boldsymbol{A}^{(l-1)} \boldsymbol{P}_{n_{l-1}}^\intercal \left( \boldsymbol{P}_{n_{l-1}} \tilde{\boldsymbol{S}}^{(l)} \right) \\ &= \tilde{\boldsymbol{S}}^{(l)\intercal} \boldsymbol{A}^{(l-1)} \tilde{\boldsymbol{S}}^{(l)} \\ &= \boldsymbol{A}^{(l)} \end{aligned}$$

## C.3 GMN

**Theorem 2** (Invariance of simplified GMN). *If the initial query network is equivariant, our simplified GMN, which employs distances rather than the Student's $t$-kernel, is invariant.*

We prove this by first proving each modified memory layer is invariant. The final result follows from the fact that the composition of an equivariant function with a series of invariant functions is invariant. Note that memory layer $l$ can be seen as a function $f_l : \mathfrak{G}' \to \mathfrak{G}'$, where $\mathfrak{G}'$ denotes the set of fully connected graphs.

Let $\mathcal{G} = (\boldsymbol{Q}^{(l-1)}, \mathbf{1}_{n_{l-1} \times n_{l-1}}) \in \mathfrak{G}'$ and $\boldsymbol{P}_{n_{l-1}}$ be a permutation matrix. Furthermore, let

$$\left(\boldsymbol{Q}^{(l)}, \mathbf{1}_{n_l \times n_l}\right) := f_l(\boldsymbol{P}_n \boldsymbol{Q}^{(l-1)}, \mathbf{1}_{n_{l-1} \times n_{l-1}})$$

$$\left(\boldsymbol{Q}^{(l)\prime}, \mathbf{1}_{n_l \times n_l}\right) := f_l(\boldsymbol{Q}^{(l-1)}, \mathbf{1}_{n_{l-1} \times n_{l-1}})$$

To achieve our goal, it suffices to show that $\boldsymbol{Q}^{(l)}$ equals $\boldsymbol{Q}^{(l)\prime}$. Let $\boldsymbol{D}_h^{(l)}$ denote the distance matrix between the queries $\boldsymbol{Q}^{(l-1)}$ and the keys in the $h$-th head of the $l$-th memory layer, then

$$\boldsymbol{Q}^{(l)} = \mathrm{ReLU}\left(\mathrm{softmax}\left(\mathrm{1x1Conv}\left(\boldsymbol{D}_1^{(l)}, \ldots, \boldsymbol{D}_H^{(l)}\right)\right)^{\mathsf{T}} \boldsymbol{Q}^{(l-1)} \boldsymbol{W}^{(l)}\right)$$

Since the distance matrix for the permuted matrix is simply the permuted distance matrix, it follows that:

$$
\begin{aligned}
\boldsymbol{Q}^{(l)\prime} &= \mathrm{ReLU}\left(\mathrm{softmax}\left(\mathrm{1x1Conv}\left(\boldsymbol{P}_{n_{l-1}}\boldsymbol{D}_1^{(l)}, \ldots, \boldsymbol{P}_{n_{l-1}}\boldsymbol{D}_H^{(l)}\right)\right)^{\mathsf{T}} \boldsymbol{P}_{n_{l-1}}\boldsymbol{Q}^{(l-1)} \boldsymbol{W}^{(l)}\right) \\
&= \mathrm{ReLU}\left(\mathrm{softmax}\left(\mathrm{1x1Conv}\left(\boldsymbol{D}_1^{(l)}, \ldots, \boldsymbol{D}_H^{(l)}\right)\right)^{\mathsf{T}} \boldsymbol{P}_{n_{l-1}}^{\mathsf{T}} \boldsymbol{P}_{n_{l-1}}\boldsymbol{Q}^{(l-1)} \boldsymbol{W}^{(l)}\right) \\
&= \mathrm{ReLU}\left(\mathrm{softmax}\left(\mathrm{1x1Conv}\left(\boldsymbol{D}_1^{(l)}, \ldots, \boldsymbol{D}_H^{(l)}\right)\right)^{\mathsf{T}} \boldsymbol{Q}^{(l-1)} \boldsymbol{W}^{(l)}\right) \\
&= \boldsymbol{Q}^{(l)}
\end{aligned}
$$

# D   Further details on Remark 2

The unsupervised loss employed to learn GMNs consists of a summation of

$$D_{\mathrm{KL}}\left(\boldsymbol{Q}^{(l)} \| \boldsymbol{P}^{(l)}\right) = \sum_{i=1}^{n_{l-1}} \sum_{j=1}^{n_l} P_{ij}^{(l)} \log \frac{P_{ij}^{(l)}}{Q_{ij}^{(l)}}$$

over all layers, where the matrix $\boldsymbol{P}^{(l)} \in \mathbb{R}^{n_{l-1} \times n_l}$ is defined such that

$$P_{ij}^{(l)} = \frac{\left(S_{ij}^{(l)}\right)^2 / \left(\sum_i S_{ij}^{(l)}\right)}{\sum_{j'}\left[\left(S_{ij'}^{(l)}\right)^2 / \left(\sum_i S_{ij'}^{(l)}\right)\right]}.$$

The intuition here is to enforce cluster purity by pushing the probabilities $P_{i:}^{(l)}$ towards their re-normalized squares. A perhaps counter-intuitive outcome of this design is that choosing $n_l$ identical keys, which results in totally uniform cluster assignments, perfectly minimizes the Kullback-Leibler divergence. To verify so, we set $Q_{ij} = 1/n_l$ for all $i = 1 \ldots n_{l-1}$ and $j = 1 \ldots n_l$, that is

$$P_{ij}^{(l)} = \frac{\frac{n_{l-1}^{-2}}{n_l n_{l-1}^{-1}}}{n_{l-1} \frac{n_{l-1}^{-2}}{n_l n_{l-1}^{-1}}} = \frac{1}{n_{l-1}} \Rightarrow D_{\mathrm{KL}}\left(\boldsymbol{Q}^{(l)} \| \boldsymbol{P}^{(l)}\right) = 0$$

# E Additional embeddings

Figures S2-S13 illustrate the activations throughout the models' main layers, similarly to Figures 5 and 6 from the main text. For better visualization, we force the embeddings (plots) to share a common aspect ratio and include the number of nodes at each layer inside brackets. These numbers correspond to the number of rows of each embedding matrix. Also, the color scale of each embedding matrix is normalized according to its own range. Therefore, color scales are not comparable across different embedding matrices. This allows us to assess the degree of homogeneity in each embedding matrix more accurately. Figures S2-S13 reinforce our findings that GNNs learn smooth signals across features at early layers. This make the pooling strategies less relevant for learning meaningful hierarchical representations.

Figure S2: GRACLUS on NCI1.

Figure S3: COMPLEMENT on NCI1.

Figure S4: GRACLUS on MOLHIV.

Figure S5: COMPLEMENT on MOLHIV.

Figure S6: GMN on NCI1.

Figure S7: Random-GMN on NCI1.

Figure S8: GMN on MOLHIV.

Figure S9: Random-GMN on MOLHIV.

Figure S10: DIFFPOOL on NCI1

Figure S11: $\mathcal{N}$-DIFFPOOL on NCI1.

Figure S12: DIFFPOOL on MOLHIV.

Figure S13: $\mathcal{N}$-DIFFPOOL on MOLHIV.

## Footnotes

[2]https://chrsmrrs.github.io/datasets/

[3]https://github.com/snap-stanford/ogb/tree/master/examples/graphproppred/mol