[Reviews · NeurIPS 2020]

Review 1

Summary and Contributions: The paper deals with supervised graph classification and regression. Specifically, it investigates the impact of pooling for learning graph-level vectorial representation based on graph neural networks. The paper argues that current (cluster-based) pooling methods do not provide benefits over simple baselines (such as random cluster assignments). The authors empirically investigate well-known cluster-based GNN pooling layers on four well-known datasets. By running a set of carefully crafted experiments, they conclude that current cluster-based pooling layers do not provide benefits over random cluster assignments and argue that is because common GNN layers suffer from "over smoothing" limiting the effectiveness of such pooling methods.

Strengths: - Clearly written - Research question raised is meaningful (has not been studied before, as far as I know) - Raises awareness for sloppy evaluation protocols inherent to most GNN papers - Provides some (but limited) insights on the failure of common pooling layers

Weaknesses: - only evaluated on *four* (mostly small) datasets with small graphs stemming from chemo- and bioinformatics - a lot of the arguments are of a handwavy nature - strong bold claims, that are only partially backup-ed with experiments

Correctness: The paper raises bold claims, but the experiment's extent does not live up to this, e.g., - only four molecular datasets are used. Nowadays, there are a large number of datasets with a varying number of sizes and structures available, e.g., ogb.stanford.ede, graphlearning.io, moleculenet.ai, ... - It is not clear why the paper does not tune the hyperparameters of the employed GNNs layers and solely resorts to hyperparameters used in the respective papers. By this, the results only hold for the specific set of hyperparameters used (performance of GNNs can be quite brittle with regard to hyperparameters).

Clarity: The paper is written clearly and is easy to follow, complemented by helpful illustrations throughout the text.

Relation to Prior Work: As far as I know, this is the first paper that tries to isolate the effectiveness of pooling for GNNs. The discussion of related work regarding pooling and GNNs seems adequate.

Reproducibility: Yes

Additional Feedback: I have a number of questions and remarks: Questions: - lines 203-203: "This happens because initial convolutional layers extract low-frequency information across the graph input channels" What does that mean formally? Please be more precise. Remarks: - lines 98-102: The complement of a graph preserves the complete structural information of the graphs, i.e., the isomorphism type of a graph is in 1-to-1 to the isomorphism type of the complement. This also holds for subgraphs. Hence, it is not surprising that clustering on the complement graph results in good results. - "oversmoothing" was only shown for the GCN layer, it might not hold for other layers, e.g., GIN. - calling fellow researcher's work "embarrassing" (l. 193) is not helpful Minor: - Ref. [25] seemed to be used arbitrarily - Ref. [43/4] seems to be the same paper, Zitao Ying and Rex Ying are the same person. ==== Post rebutal ==== I have slightly increased my score. However, I still believe that the authors should conduct more experiments on larger non-molecule data, and conduct an extensive hyperparameter search.


Review 2

Summary and Contributions: This work studies pooling operations in graph neural networks and concludes that existing pooling techniques do not improve performances.

Strengths: It is an interesting rigorous study of graph pooling operations.

Weaknesses: One concern is that the evaluation of local pooling techniques is done in the context of graph-level tasks, i.e. graph regression for e.g. ZINC and graph classification for e.g. MNIST. Local pooling techniques should have also been evaluated in the context of node/edge-level tasks like node classification or link prediction. Graph pooling techniques may provide no improvement or even decrease the performances on small graphs, s.a graphs with 20-60 nodes used in this work. However, the question remains open for large graphs with 100K-1M nodes. This is in comparison with imagenet images of grid size 256x256 pixels = graph with 65k nodes, where pooling or strided convolutions improve performances. It would interesting to run some experiments on the recent large OGB datasets to confirm that these results are not caused by the small graph sizes.

Correctness: Likely.

Clarity: Yes

Relation to Prior Work: Yes

Reproducibility: Yes

Additional Feedback:


Review 3

Summary and Contributions: This paper analyzes the effectiveness of local pooling layers in existing graph neural networks for the task of graph representation learning. By analyzing three existing popular local pooling methods, the authors show that the pooling layers are not responsible for the success of GNNs, which are mainly due to the graph convolutional layers. This

Strengths: + This paper studies an important problem, the effectiveness of local pooling layers in existing graph neural networks + The conclusions are very instrumental, which sheds new light onto the role of pooling.

Weaknesses: - the analysis are only limited to three local pooling methods

Correctness: Yes

Clarity: Yes, well written and easy to read

Relation to Prior Work: Yes

Reproducibility: Yes

Additional Feedback: Instead of designing new pooling methods for whole graph representation learning, this paper instead analyzes the effectiveness of existing local pooling methods, which have been attracting growing interests recently. The authors show that the local pooling layers are actually not responsible for the success of GNNs by randomizing the cluster assignments or clustering on the complementary graph. This is a really interesting and important paper. There are indeed increasing concerns on the effectiveness of recently proposed different pooling techniques, which are more due to different evaluation configurations. For example, the following paper raises a similar concern: Federico Errica, Marco Podda, Davide Bacciu, Alessio Micheli et al. A Fair Comparison of Graph Neural Networks for Graph Classification. A minor concern on the experiments is that the analysis would be more convincing by considering more pooling techniques (e.g. SortPool). Another question is how would the number of graph convolutional layers affect the conclusion?


Review 4

Summary and Contributions: This paper challenges the common assumption that local pooling of nodes in Graph Neural Networks is beneficial to improve performances. In the experiments, the authors show that: - using pooling layers does not lead to an increase in performance with respect to simple baselines such as a) pooling disconnected nodes or b) pooling random node subsets; - pooling learns approximately homogeneous features across nodes (which corroborates the first finding); - using unsupervised losses to force nodes to create pure clusters has little to no impact on predictive performances; - local node pooling performs similarly to global node pooling. ------ Post feedback update: In light of the feedback received from the authors, which in my opinion responds to all the concerns expressed by me in a satisfactory way, I have decided raise this paper's score to 7, and I recommend acceptance.

Strengths: This paper provides evidence that pooling in GNNs might not be of such influence to achieve good results (even if there are other reasons to consider pooling beyond performance evaluation). Anyway, even though it is a harsh claim, it seems supported by empirical evidence. I believe this paper might be of use to the graph learning community, as it establishes novel standards to satisfy, and useful baselines to outperform, in order to assess the validity and usefulness of novel pooling methods.

Weaknesses: 1) The work is incremental, nothing really new is proposed (expecially in term of solutions). 2) I am not fully convinced that the argument of the paper holds for GRACLUS for two reasons: - It is well known that some problems in a graph have a dual problem in the complement graph (e.g. clique vs. independent set). Thus, it might be that the fact the model learned something about the dual problem in the complement graph is sufficient to explain the similar performances. - The authors do not provide strikingly convincing evidence that pooling in GRACLUS learns homogeneous representations (as opposed to the other methods where the evidence is clear) However, I am willing to increase my judgement if the authors are able to convince me in the rebuttal phase.

Correctness: Overall, the empirical methodology seems correct. However, I have some key observations to make: - figure 5 and 6 are fairly different one to the other, even though they are supposed to show the same aspect (homogeneous node representations) from two different points of view (non-local in figure 5, local in figure 6). I expect the two figures to be "equalized" using the same 5 graphs used in figure 5. It would be also convenient to have more examples of homogeneous node representations in the appendix, possibly for each of the adopted datasets. I believe this is necessary since this is the main explanation the authors give to the trend they are highlighting. - one of the datasets used in this study is IMDB-B. However, [1] showed that it is affected by a serious isomorphism bias (approximately 50% of the graphs are isomorphic, some of them with different labels). As such, the results on this dataset are not reliable. I suggest to use the "cleaned" version of IMDB-B, or alternatively to drop it altogether and use another social dataset to prove their point. [1] arxiv.org/abs/1910.12091

Clarity: The paper is well written, well structured, and pleasing to read.

Relation to Prior Work: Given the space constraints, I find prior work properly discussed.

Reproducibility: Yes

Additional Feedback: on line 115, the score of COMPLEMENT on IMDB-B is 69.4+/-0.7, while in Table 1 it is 70.6+/-5.1 line 193: please use a more polite tone ("if not embarassing") line 238: perceptive -> perspective

[Author Response · NeurIPS 2020]



Figure 1: **Upper**: GRACLUS; **Lower**: COMPLEMENT.

Figure 2: Numbers denote the Graclus performance on ZINC. Colors indicate the performance gap between Graclus and Complement. Blue tones denote cases on which Complement obtains lower MAE.

Table 1: Avg. results (AUC for molhiv and Accuracy for others) for the cleaned version of IMDB and two additional larger datasets.

|  | IMDB-B | DD | molhiv |
|---|---|---|---|
| GRACLUS | 70.5 | 72.1 | 75.2 |
| COMPLEMENT | 70.1 | 72.0 | 74.7 |
| DIFFPOOL | 70.9 | 77.0 | 70.5 |
| $\mathcal{U}$ DIFFPOOL | 70.2 | 77.5 | 71.0 |
| GMN | 68.4 | 74.5 | 73.9 |
| Rand. GMN | 70.2 | 73.6 | 74.1 |
| Dist. GMN | 70.0 | 74.9 | 73.9 |
| Mincut | 70.4 | 76.8 | 71.7 |
| $\mathcal{N}$-Mincut | 71.0 | 76.7 | 73.5 |

We thank the reviewers for their valuable feedback. We are glad the reviewers found we are addressing an important problem (**R1**, **R4**) for which we carried a rigorous study (**R3**) supported by evidence (**R5**) and that our work sheds new light onto the role of pooling (**R4**). We have made our best effort to address all the questions given the limited space.

**@R1 @R5 Difference between GRACLUS and COMPLEMENT.** Thank you for raising this issue. We recognize that our explanation might mislead the reader to believe that COMPLEMENT fully operates on complement graphs. In fact, we only employ the **complement graph to compute cluster assignments** $S^{(l)}$. With the assignments in hand, we apply the **pooling operation** (Eqs. 2 and 3) **using the original graph structure** $(A^{(l)}, X^{(l)})$. That being said, there is no clear reason to believe there is a 1-to-1 correspondence between the representations achieved by GRACLUS and COMPLEMENT. As an example, the Figure 1 shows how different these coarsened structures can be. Note that the paired representations are not complementary. We will improve the description of COMPLEMENT in the manuscript.

**@R1 @R3 Additional experiments with larger datasets and graphs.** We initially considered datasets that have been broadly used to validate novel pooling methods. Importantly, we have also strived to avoid common yet misleading evaluation protocols and datasets recently reported in benchmarking papers. For completeness, we have also run new experiments (Table 1) on a dataset with **larger graphs (DD, $\approx 250$ nodes)** and an **OGB dataset (molhiv, $\approx 50$K samples)**. These additional results **reinforce our initial findings**. We now have results for a total of eight datasets to support our claims. We will include these new results in the appendix and briefly discuss them in the main paper.

**@R1 "Please be more precise" (about low-frequency graph filters).** Agreed! We tried to convey that, in analogy with 2D convolutions, $d$-dimensional node features of a graph correspond to $d$ image channels. And the filtering operation (convolution) acts on signals defined over the nodes. We will rewrite the sentence more precisely.

**@R1 "...results only hold for the specific set of hyperparameters".** We have followed general guidelines from the original (or benchmarking) papers and used the same hyperparameters for our variants. We believe that this methodology promotes a fair assessment of the role of local pooling. Also, we have shown results for four methods and (now) eight datasets to support our main claims. As additional evidence, Figure 2 reports the performance from Graclus and Complement for different hyperparameter settings, for all of which the performance gap is < 0.05 MAE.

**@R1 "oversmoothing was only shown for the GCN layer...".** Note that we only employ GCN layers (i.e., mean-based operators) in DiffPool. For the remaining methods, we adopted sum-based operators, which are known to be more expressive than mean-based ones. Thus, **we believe our results are not limited to GCN convolutions**.

**@R4 Effect of more/less convolutions.** We do not expect local pooling to be more effective with more convolutions. As an example, Figure 2 shows the performance gap between Graclus and Complement as a function of the number of convolutions. Both models obtain very similar performance as we increase the number of layers.

**@R4 "A minor concern ... considering more pooling techniques".** We also provide **results for MinCutPool (ICML 2020) in Appendix B**, which corroborates our findings. We will mention this in the main text.

**@R5 @R1 "Please use a more polite tone".** Good point; we fully agree and will remove that slip.

**@R5 Removing isomorphisms from IMDB.** Thanks for pointing this out. We have rerun the experiments with the cleaned version of the dataset (see Table 1). Gladly, this change **had no impact on our conclusions**.

**@R5 "...more representations in the appendix".** We agree that this will add illustrative value to our paper. We have saved a number of these representations from which we have chosen only a few to illustrate our claims. We will add more embeddings to the Appendix and also adjust the Figures 5 and 6 to make a direct comparison easier.

**@R5 Are Graclus representations smooth?** We agree that the representations learned by Graclus are not as smooth as those from GMN and DiffPool. However, Complement produces much smoother representations while maintaining the same performance. As an example, for ZINC graphs, we computed the avg std deviation of their embeddings before and after the 1st pooling layer, for which Complement achieves one order of magnitude lower compared to Graclus.

[Meta-Review · NeurIPS 2020]

All the reviewers have supported the acceptance of this paper.